# SSA-VMD for UWB Radar Sensor Vital Sign Extraction

**DOI:** 10.3390/s23020756

**Published:** 2023-01-09

**Authors:** Huimin Yu, Wenjun Huang, Baoqiang Du

**Affiliations:** College of Information Science and Engineering, Hunan Normal University, Changsha 410081, China

**Keywords:** vital sign monitoring, human detection, ultra-wideband sensor, smart grid

## Abstract

The combination of advanced radar sensor technology and smart grid has broad prospects. It is meaningful to monitor the respiration and heartbeat of grid employees under resting state through radar sensors to ensure that they are in a healthy working state. Ultra-wideband (UWB) radar sensor is suitable for this application because of its strong penetration ability, high range resolution and low average power consumption. However, due to weak heartbeat amplitude and measurement noise, the accurate measurement of the target heart rate is a challenge. In this paper, singular spectrum analysis (SSA) is proposed to reconstruct the eigenvalues of noisy vital signs to eliminate noise peaks around the heartbeat rate; combined with the variational modal decomposition (VMD), the target vital signs can be extracted with high accuracy. The experiment confirmed that the target vital sign information can be extracted with high accuracy from ten subjects at different distances, which can play an important role in short distance human detection and vital sign monitoring.

## 1. Introduction

The combination of radar technology and the smart grid system has broad application prospects. The traditional inspection of the artificial power grid will threaten the life and health of inspectors in extreme weather such as thunderstorms. In today’s smart grid system, the airborne laser radar has been used for power patrol, and the relevant information of power lines is automatically extracted through the laser point cloud data [1], which not only ensures the safety of staff, but also saves human resources. Due to frequent typhoons in coastal areas, they have a great impact on the stable operation of the steel tower of the transmission network. Once the steel tower of the transmission network inclines, it will endanger the security of the power grid. Tilt measurement based on radar sensor has been put into use in China Southern Power Grid. Although the above radar technology has been applied to the smart grid system to ensure the stable and safe operation of the grid. However, continuous health monitoring of staff cannot be ignored. Various physiological indicators of the human body, such as respiration, heartbeat, blood pressure, blood oxygen, etc., can reflect the physical condition of the human body in real time [2,3,4]. It is meaningful to find abnormal vital signs in time to ensure that they are in a healthy working state [5].

With the development of wireless communication technology, the extraction of human respiration and heartbeat has entered the wireless era. Compared with the traditional contact vital sign monitoring technology, it can measure the target vital signs without knowing and comfort. Reference [6] proposed to implement respiratory monitoring based on the received signal strength (RSS) of Wi-fi. Due to the influence of the indoor environment multipath, the signal strength fluctuates, and the stability and accuracy of this method will be affected. The non-contact BCG can achieve heart rate (HR) detection by capturing a video with imaging equipment, such as collecting head feature points for tracking [7]. However, the head movement in BCG is weak and is mixed with other non-autonomous head movements, resulting in a low SNR of the BCG signal extracted from the head movement, which affects the accuracy of HR. Due to its strong penetration ability, good target recognition ability, high range resolution, low average power consumption and other advantages [8], UWB radar has been applied in many fields, including mobile target detection [9], wall through imaging [10], rescue [11], indoor target positioning [12] and public security [13].

The original vital signs signal cannot be directly obtained from the radar echo. Because there are direct waves reflected by static targets and they may have DC components in the radar echo, data preprocessing is required for the radar echo first. For the original vital sign signal, reference [14] proposed that ensemble empirical mode decomposition (EEMD) was used to decompose the vital signs, but the EEMD would have the problem of a false mode. In recent years, references [15,16,17,18] proposed to achieve relatively complete separation of human respiratory and heartbeat signals based on variational modal decomposition (VMD). VMD is a new adaptive decomposition method based on Wiener filter and Hilbert transform [19]. It can adaptively match the modal bandwidth and the optimal center frequency in the decomposition process, and has more solid mathematical theoretical support than the empirical mode decomposition (EMD) algorithm. Singular spectrum analysis (SSA) is a powerful method for studying nonlinear time series data and has applications in the field of mechanical fault diagnosis and prediction [20]. It has been applied to the field of vital sign monitoring in recent years. References [21,22] proposed to use SSA to decompose the vital sign, and successfully separated respiration and heartbeat by clustering the singular values. However, due to the weak heartbeat amplitude, the impact of the noise caused by the measurement environment or hardware cannot be ignored [23], the intermodulation of heartbeat frequency peak and noise will cause errors, which cannot meet the requirements of high accuracy. 

In this paper, SSA is innovatively applied to reduce the noise of vital signs and eliminate noise interference, combined with the VMD algorithm to achieve high accuracy of target vital sign detection based on a UWB radar sensor. The main contributions are as follows: (1) Singular value decomposition (SVD) is proposed to adaptively implement UWB radar echo to eliminate static clutter and reduce echo noise. (2) Singular spectrum analysis (SSA) is proposed to eliminate noise peaks around the heartbeat rate. The experiment confirmed that the target vital sign information can be extracted with high accuracy from ten subjects at different distances, which can play an important role in human detection and vital sign monitoring.

## 2. UWB Radar Vital Sign Signal Detection Model

It is common knowledge that when the human body breathes, the chest cavity will continuously fluctuate periodically. However, the beating of the heart can also make the chest rise and fall slightly. By transmitting electromagnetic waves, the radar can detect the ups and downs on the chest caused by the breathing and heartbeat. It is mainly reflected in the changing distance between the antenna of the radar receiver and the chest cavity. The periodic change of this distance reflects the frequency of respiration and heartbeat. Therefore, non-contact vital signs can be extracted through the radar, as shown in Figure 1. In a radar echo at a certain distance, assuming that the instantaneous distance of the radar antenna to the human thorax is:(1)$$d(t)=d_{0}+A_{r}\sin(2\pi f_{r}t)+A_{h}\sin(2\pi f_{h}t),$$
where $d_{0}$ is average distance from the human thorax to the antenna, $A_{r}$ and $A_{h}$ are the amplitudes of the respiration and heartbeat signal, $f_{r}$ and $f_{h}$ are the frequency of the respiration and heartbeat signal. The delay between the transmitter pulse and receiver pulse is:(2)$$\tau_{v}(t)=\frac{2d(t)}{v},$$
where v is the speed of light. In the tested area, ideally only the human thoracic cavity is moving, and the others are static. Therefore, radar pulse response is the sum of human thorax response and environmental response:(3)$$h(\tau ,t)=a_{v}\delta (\tau -\tau_{v}(t))+\sum_{i}a_{i}\delta (\tau -\delta_{i}),$$
where $a_{v}$ is the reflection coefficient of the human thorax, $a_{i}$ is the reflection coefficient of different targets in the environment and δ is the unit impulse response. The ideal reception signal can be expressed as:(4)$$r(\tau ,t)=s(\tau )*h(\tau ,t)=a_{v}s(\tau -\tau_{v}(t))+\sum_{i}a_{i}s(\tau ,\tau_{i}),$$
where s(τ) is the first-order Gaussian pulse signal.

The echo signal after data preprocessing can be expressed as:(5)$$r(\tau ,t)=a_{v}s(\tau -\tau_{v}(t)),$$

With the two-dimensional Fourier transform for radar fast-time τ and slow-time t, the radar echo signal spectrum can be expressed as:(6)$$Y(f,v)=\int_{-\infty }^{+\infty }\int_{-\infty }^{+\infty }a_{v}s(\tau -\tau_{v}(t))e^{-j2\pi ft}e^{-j2\pi v\tau }=a_{v}S(v)\int_{-\infty }^{+\infty }e^{-j2\pi ft}e^{-j2\pi v\tau_{v}(t)}dt,$$
where S(v) is the result of the Fourier transform of s(τ). Convert the above formula to a Bessel function:(7)$$Y(f,v)=a_{v}S(v)e^{-j2\pi v\tau_{0}}\int_{-\infty }^{+\infty }\left(\sum_{k=-\infty }^{\infty }J_{k}(\beta_{r}v)e^{-j2\pi kf_{r}t})\times (\sum_{k=-\infty }^{\infty }J_{l}(\beta_{h}v)e^{-j2\pi kf_{h}t}\right)e^{-j2\pi ft}dt,$$
where $\beta_{r}$ is $2\pi A_{r}$, $\beta_{h}$ is $2\pi A_{h}$. General vital signs signal detection algorithms only take Fourier transforms for radar slow-time:(8)$$Y(f,\tau )=\int_{-\infty }^{+\infty }Y(f,v)e^{j2\pi v\tau }dv=a_{v}\sum_{k=-\infty }^{\infty }\sum_{k=-\infty }^{\infty }G_{kl}(\tau )\delta (f-kf_{r}-lf_{h}),$$
(9)$$G_{kl}(\tau )=\int_{-\infty }^{+\infty }S(v)J_{k}(\beta_{r}v)J_{l}(\beta_{h}v)e^{j2\pi v(\tau -\tau_{0})}dv,$$

In summary, fundamental and harmonics of respiration and heartbeat exist in the radar slow-time signal spectrum: $kf_{r}$ and $lf_{h}$.

For Equation (5), we discretize radar fast-time and slow-time, $\tau =m\delta_{T}$, $t=nT_{s}$:(10)$$R[m,n]=a_{v}s(m\delta_{T}-\tau_{v}(nT_{s})),$$
where $T_{s}$ is slow-time sampling interval, n=0,1,2…,N−1, $\delta_{T}$ is the fast-time sampling interval, m=0,1,2…,M−1. Original vital sign signals were sampled and stored in the m×n size radar echo matrix.

## 3. Radar Echo Preprocessing Process

In the actual detection environment, the radar echo is more complex. In addition to the static clutter and vital signs mentioned above, it may contain a direct-current (DC) component. The appearance of the DC component is related to the base drift of the working temperature of the radar receiver hardware. Because both DC component and static clutter are constant, they can be removed by the same method. In order to extract the one-dimensional time domain vital signs, the preprocessing of radar echo is essential. Figure 2 shows the radar echo preprocessing process.

### 3.1. Clutter Suppression (SVD)

Singular Value Decomposition (SVD) is an algorithm widely used in the field of machine learning. It is often used for matrix data compression, that is, to express the original matrix by factorization. Because the radar echo can be regarded as a two-dimensional matrix based on distance and time, SVD can be used to reduce its dimensionality—that is, to eliminate static clutter and reduce echo noise—and retain the original vital sign signals captured in the radar echo. The radar echo is decomposed by SVD:(11)$$R=U\cdot \Lambda \cdot V^{H},R\in A^{K\times N},$$
where R is the radar echo, U is unitary matrix of order K×K, $V^{H}$ is unitary matrix of order N×N, Λ is the diagonal matrix of A×A,A=min{K,N} where the singular values are located. The singular values are arranged in the diagonal matrix Λ from largest to smallest:(12)$$\Lambda =\left[\begin{array}{cccc} \Lambda_{11} & 0 & \ldots & 0\\ 0 & \Lambda_{22} & \ldots & \ldots \\ \ldots & 0 & \ldots & 0\\ 0 & \ldots & 0 & \Lambda_{AA} \end{array}\right],\Lambda_{11}>\Lambda_{22}>\ldots >\Lambda_{AA}$$

The singular values obtained by SVD correspond to the components in the radar echo matrix. Singular values have the following characteristics:

Static clutter and DC correspond to a large singular value, usually only one, which is $\Lambda_{11}$.The singular value corresponding to the matrix noise is small, but the number is large, and is distributed in the lower right corner of the diagonal matrix.The size of the singular value corresponding to the vital signs signal is obviously separate from the static clutter and matrix noise, and it is distributed several times after the static clutter.

Generally, the second singular value $\Lambda_{22}$ in the diagonal matrix is selected for radar echo reconstruction, and the radar echo with static clutter and matrix noise removed can be obtained by:

Zero the singular values in the diagonal matrix Λ except $\Lambda_{22}$, keeping its positions in the original matrix Λ unchanged:(13)$$\Lambda^{\prime }=\left[\begin{array}{ccccccc} 0 & & & & & & \\ & \Lambda_{22} & & & & & \\ & & 0_{} & & & & \\ & & & \ldots & & & \\ & & & & 0_{} & & \\ & & & & & \ldots & \\ & & & & & & 0 \end{array}\right]$$
(14)$$R^{\prime }=U\cdot \Lambda^{\prime }\cdot V^{H},R^{\prime }\in A^{K\times N},$$
where $R^{\prime }$ is the radar echo reconstructed based on SVD.

### 3.2. Maximum Distance Gate Selection

After filtering out static clutter and preliminary noise reduction in the matrix by SVD, the human vital signs signal can be considered to be the most energetic in the radar echo. The distance from the thorax to the radar antenna fluctuates around $d_{0}$ due to thoracic ups and downs. By selecting the most energetic distance gate from the radar distance dimension, we can get a slice of data about radar slow-time—that is, the maximum distance gate selection.
(15)$$E_{m}=\sum_{i=1}^{N}R_{m,i}^{2},$$
(16)$$\hat{m}=\arg\max_{m}(E_{m}),$$
(17)$$s(i)=R_{\hat{m},i},$$
where s(i) is the original vital signs signal, and its size is 1×n. It is the superimposed signal of respiration and its harmonics, heartbeat signal and noise.

## 4. Vital Sign Processing Method

### 4.1. Singular Spectrum Analysis (SSA) 

The analysis object of this algorithm is a one-dimensional time series, and the trajectory matrix is first constructed by lagging the original series and performing SVD on it. Then, the additive components of the signal are separated and divided into reconstructed original sequences according to the singular value size selection. The whole process is summarized in four parts: embedding, SVD decomposition, grouping and reconstruction.

Embedding

Assuming that the object of analysis is a one-dimensional finite time series: [$x_{1},x_{2},\cdot \cdot \cdot ,x_{N}$], N is the length of the data. First, choose a suitable window length L. The data sequence is lag-arranged according to the length of L to obtain a trajectory matrix with L rows and N−L+1 columns.
(18)$$\left[\begin{array}{cccc} x_{1} & x_{2} & \cdot \cdot \cdot & x_{N-L+1}\\ x_{2} & x_{3} & \cdot \cdot \cdot & x_{N-L+2}\\ \cdot \cdot \cdot & \cdot \cdot \cdot & & \cdot \cdot \cdot \\ x_{L} & x_{L+1} & \cdot \cdot \cdot & x_{N} \end{array}\right]$$

Set N−L+1 to be K. Equation (18) is the Hankel matrix of L×K.

SVD decomposition

Singular value decomposition of the trajectory matrix X:(19)$$X=\sum_{i=1}^{d}U_{i}\lambda_{i}V_{i}^{T},$$
where d is the number of non-zero singular values, $d=rank(X)\leq \min(L,K),\lambda_{1},\lambda_{2},\cdot \cdot \cdot ,\lambda_{d}$ is the singular value in descending order, $U_{i}$ and $V_{i}$ are the left and right singular vectors, respectively.

Grouping

The grouping is the representation of the trajectory matrix X is constructed from the original sequence as the sum of useful signals S and E, i.e., X=S+E. Usually, the r singular values arranged in front are artificially considered as useful components, and the d−r ones behind are considered as noisy components.

Reconstruction

First calculate the projection of the hysteresis sequence $X_{i}$ onto $U_{m}$:(20)$$a_{i}^{m}=X_{i}U_{m}=\sum_{j=1}^{L}x_{i+j}U_{m},0\leq i\leq N-L,$$

$X_{i}$ denotes the i-th column of the trajectory matrix X, $a_{i}^{m}$ is the weight of the time-evolving type reflected by $X_{i}$ at time slot $x_{i+1},x_{i+2,}\cdot \cdot \cdot ,x_{i+L}$ of the original sequence, called the temporal principal component (TPC). The reconstructed sequence can be calculated via the following equation:(21)$$x_{i}^{k}=\left\{\begin{array}{l} \frac{1}{i}\sum_{j=1}^{i}a_{i-j}^{k}U_{k,j},1\leq i\leq L-1\\ \frac{1}{L}\sum_{j=1}^{L}a_{i-j}^{k}U_{k,j},L\leq i\leq N-L+1\\ \frac{1}{N-L+1}\sum_{j=i-N+L}^{L}a_{i-j}^{k}U_{k,j},N-L+2\leq i\leq N \end{array}\right\}$$

The window length L is an important parameter of singular spectrum analysis. The larger the window length, the finer the sequence decomposition. However, because the window length L and the singular value decomposition of the trajectory matrix K=N−L+1 are symmetric, the window length is usually less than N/2. 

### 4.2. Variational Modal Decomposition (VMD)

Variational modal decomposition can decompose the original signal into a set of intrinsic mode functions (IMF) components with sparse characteristics by searching for the optimal solution of the constrained variational model. VMD is written as a constrained variational problem [19]:(22)$$\underset{u_{k},\omega_{k}}{\min}\left\{{\left\|\sum_{k}u_{k}(t)-f(t)\right\|}_{2}^{2}+\alpha \sum_{k}{\left\|\partial t\left[u_{k}e^{-jw_{k}t}\right]\right\|}_{2}^{2}\right\},$$
where k is the number of IMFs and f is the input signal, $\left\{u_{k}\right\}=\left\{u_{1},u_{2},\ldots ,u_{k}\right\}$ and $\left\{\omega_{k}\right\}=\left\{\omega_{1},\omega_{2},\ldots ,\omega_{k}\right\}$ are all modes and their center frequencies, respectively. The above equation is solved by introducing quadratic penalty functions and Lagrange multipliers:(23)$$J={\int_{0}^{\infty }\alpha \sum_{k}{\left(\omega -\omega_{k}\right)}^{2}{\left(u_{k}(w)\right)}^{2}+\left(\sum_{k}u_{k}(\omega )-\hat{f}(w)\right)}^{2}+\hat{\lambda }(w)\left(\hat{f}(w)-\sum_{k}u_{k}(w)\right),$$
where α denotes the equilibrium parameter of the data fidelity constraint. Then, the variational problem is solved by the alternating direction multiplier algorithm, and all modal components are obtained from the solution in the spectral domain as follows:(24)$$u_{k}(\omega )=\frac{\hat{f}(w)-\sum_{i\neq k}u_{i}(w)+\frac{\hat{\lambda }(w)}{2}}{1+\alpha {\left(\omega -\omega_{k}\right)}^{2}},$$
(25)$$\omega_{k}=\frac{\int_{0}^{\infty }\omega {\left|u_{k}(w)\right|}^{2}d\omega }{\int_{0}^{\infty }{\left|u_{k}(w)\right|}^{2}d\omega }$$
(26)$${\hat{\lambda }}^{n+1}(\omega )={\hat{\lambda }}^{n}(\omega )-\tau \left(\hat{f}(w)-\sum_{k}u_{k}(w)\right),$$
where $\omega_{k}$ is calculated at the center of gravity of the power spectrum of the corresponding mode and $\hat{\lambda }$ is updated using gradient descent. If convergence is satisfied, the cycle stops and IMF is obtained. If convergence is not satisfied, repeat the above steps.

## 5. Experimental Results and Analysis

### 5.1. Experimental Environment

In this chapter, we used radar to real test, and simulate that employees are in a sitting position and monitor their vital signs. We also analyzed the method and effect of radar echo preprocessing based on SVD proposed in this paper. The SSA-VMD method proposed in this paper is used to compare the extraction results of the vital sign signals extracted from the echoes with VMD [15,16,17,18], reflecting the advantages of the accuracy of this method. 

The Xethru X4M03 UWB radar module from Novelda was used to acquire human vital sign signals, and the subject sat quietly at about 0.6 m in front of the radar to simulate the employee’s attitude in a resting state and wore a contact finger clip measuring device with Chinese medical certification to acquire vital sign data as the reference standard value. The X4M03 module and the experimental scenario are shown in Figure 3, and the X4M03 radar parameters are shown in Table 1.

### 5.2. Target Vital Sign Signal Extraction

Import the original radar echo collected by the upper computer software Xethru Explorer into Matlab2018 and draw the energy map; static clutter exists at close proximity and significant undulating movements at a distance of 0.6 m, as shown in Figure 4. Figure 5 shows the singular value spectrum of the original radar echo in the singular value spectrum; the singular values representing vital signs in this experiment are the second to fourth. The reason is that the chest of the human body has a certain thickness, and some electromagnetic waves pass through the front chest and then reflect. The second singular value mainly represents the reflection of electromagnetic waves from the human chest; by selecting the second singular value of the echo for reconstruction, the fluctuation of the human thorax can be seen more clearly. Next, selecting the strongest vital signs at the distance around 0.6 m (maximum distance gate selection), the slow-time data slice is the original vital signs signal which contains the respiration signal, heartbeat signal and noise, as shown in Figure 6.

In this test, the contact finger clip measuring instrument synchronously detected that the human respiration was 16 times/min (about 0.267 Hz) and the heart rate was 90 times/min (about 1.5 Hz). In the spectrum of the sign signal, there were obvious peaks at 0.2652, 0.5304, 1.014 and 1.513 Hz, which were known as respiratory fundamental frequency, respiratory second harmonic frequency, respiratory fourth harmonic frequency and heartbeat frequency, respectively. It can be seen that there was a large amount of noise around the peak heartbeat rate, which seriously interferes with the accurate extraction of the heartbeat.

### 5.3. Comparison of VMD and SSA-VMD Results

VMD decomposition of the original vital sign signal is performed to extract the sign information. Ideally, the signals to be extracted are respiratory and heartbeat signals, and the modal number k of the VMD algorithm is set to 3 (including a residual). However, due to the complex components of the measured sign signals, which contain respiratory harmonics and other disturbances stronger than the heartbeat signals, the parameters of the VMD algorithm are shown in Table 2. 

In addition to parameter k, a is also an important parameter of the VMD algorithm. a determines the bandwidth of the IMF. The larger the bandwidth, the greater the bandwidth of each IMF. Excessive bandwidth will make some components contain other component signals. Considering that the vital signs have filtered out much of the noise by SSA, we set it to a smaller value of 1000. In addition, the smaller a value can better separate the respiratory harmonic and heartbeat frequency in the heartbeat band.

As shown in Figure 7, the heartbeat frequency extracted by the VMD method is 1.576 Hz, which is without vital sign noise reduction, and the heartbeat frequency is intermodulated with noise, deviating from the standard heart rate and causing a mistakenly large error (about 5 times/min). IMF2 and IMF3 are the fourth and second harmonics of respiration, respectively, and the respiratory signal is fixed at IMF4. The respiratory frequency extracted by VMD is close to the reference standard respiratory frequency.

Next, perform the SSA-VMD proposed in this paper to the original vital sign. According to the radar parameter sampling duration and sampling rate, as well as the SSA window length L selection principle, set the window length L=400 for constructing the trajectory matrix. SVD is performed on the trajectory matrix constructed from the original vital sign, and the singular value spectrum is shown in Figure 8. It can be seen from Figure 8 that the third singular value drops precipitously, and it can be considered that the first and second singular values represent respiratory signals; the third singular value starts to characterize other components in vital signs, such as multiple harmonics of respiration and intermodulation wave. For the selection of the number of reconstructed singular values r, singular values representing heartbeat must be included, otherwise vital sign monitoring will fail. In the singular value spectrum, it is generally considered that the singular value close to zero is noise, and the singular value representing the heartbeat signal is usually slightly larger than the noise singular value. Therefore, a singular value arranged in the front and close to zero is selected as the critical value, and singular values arranged behind it are considered as noise, which can filter out the noise in vital signs to the greatest extent. After analyzing the singular value spectrum of a large number of vital signs collected in this study, the critical value for filtering the noise singular value is related to the window length L, the singular value at the 0.1×L position realizes the whole process of singular values descending from large to near zero, and it is suitable as a critical value.

Therefore, the first 40 singular values are selected for vital sign reconstruction. The time-frequency domain of the reconstructed signal is shown in Figure 9. From the perspective of time domain, dark burrs caused by measurement noise are eliminated and the SNR is enhanced. From the frequency domain perspective, the random noise around the heartbeat peak is eliminated, making subsequent heartbeat peaks easier to detect, and the frequency offset effect caused by noise is eliminated. VMD decomposition of reconstructed vital signs is performed, as shown in Figure 10. Compared with VMD, the frequency spectrum of the heartbeat signal is simpler; without frequency offset, the SSA-VMD-based method is more accurate in extracting heartbeat rate.

### 5.4. Heartbeat Rate Accuracy Comparison at Different Distances

In order not to lose generality, this chapter conducts a comparison experiment on the accuracy of heartbeat frequency extraction based on the SSA-VMD method and the VMD method at different distances. The experimental subjects were ten healthy young people, all aged 22–24 years old. The distances were set to 0.6 m and 1.2 m, and five subjects were tested at each distance. The subjects sat quietly in front of the radar and wore finger clip detectors to obtain standard physical sign information for comparison with the radar data.

The results of the experiment are shown in Table 3 and Table 4. At 0.6 m, the mean error of SSA-VMD heartbeat extraction accuracy was 0.045 Hz (less than 3 beats/min) and the average error of VMD heartbeat extraction accuracy was 0.115 Hz (about 7 beats/min). At 1.2 m, the average error of SSA-VMD heartbeat extraction accuracy was 0.041 Hz (less than 3 beats/min), and the average error of VMD heartbeat extraction accuracy was 0.074 Hz (about 4 beats/min). It can be seen from the experiments of ten subjects that the SSA-VMD-based method has a smaller error of about 0.04 Hz in heart rate detection at different distances, which is more accurate. In contrast, the VMD error is larger compared to SSA-VMD, and the average error at different distances also differs, the main reason for the difference is that the random noise undulation in the sign signal is not eliminated and there is a certain randomness.

## 6. Conclusions

In this paper, based on the detection of human vital signs by ultra-wideband radar, the SSA-VMD vital sign detection method is proposed for the existence of noise in the original vital sign signal, which seriously affects the accurate extraction of faint heartbeat frequency. The results are verified by actual detection findings as follows:(1)The experimental analysis of the actual detected signals shows that the heartbeat frequencies extracted based on SSA-VMD benefit from the noise reduction of the original sign signals, and their spectral components are single and far less affected by noise.(2)The experimental data of multiple people under different distances of actual detection show that the accuracy of the heartbeat frequency extracted based on SSA-VMD is higher than that of the single VMD method, contributing to a more accurate heartbeat frequency extraction.

## Figures and Tables

**Figure 1 sensors-23-00756-f001:**
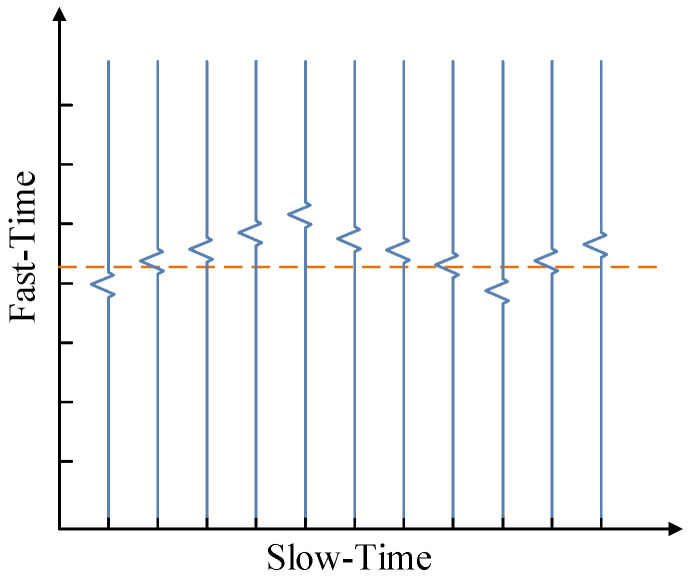
Schematic diagram of the radar echo signal.

**Figure 2 sensors-23-00756-f002:**
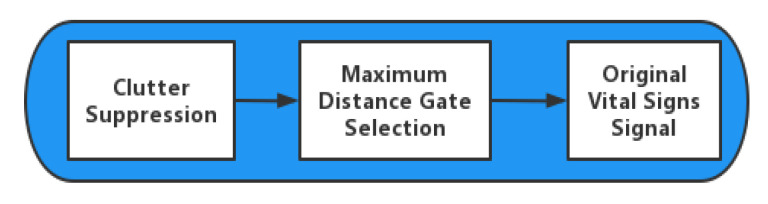
Radar echo preprocessing process.

**Figure 3 sensors-23-00756-f003:**
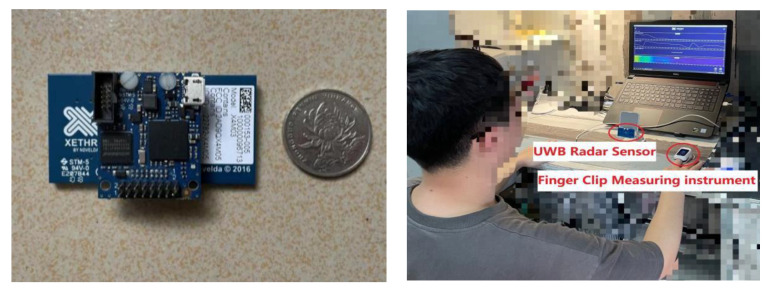
X4M03 module and experiment environment.

**Figure 4 sensors-23-00756-f004:**
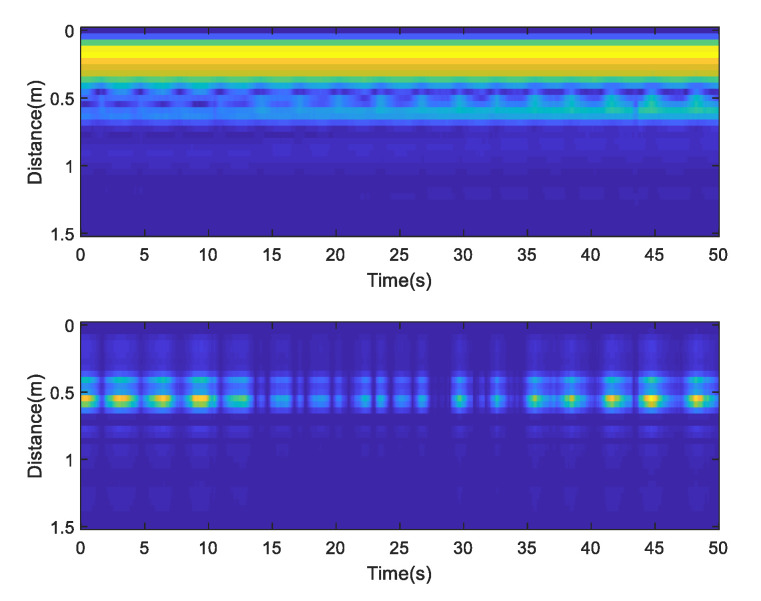
Before and after radar echo reconstruction.

**Figure 5 sensors-23-00756-f005:**
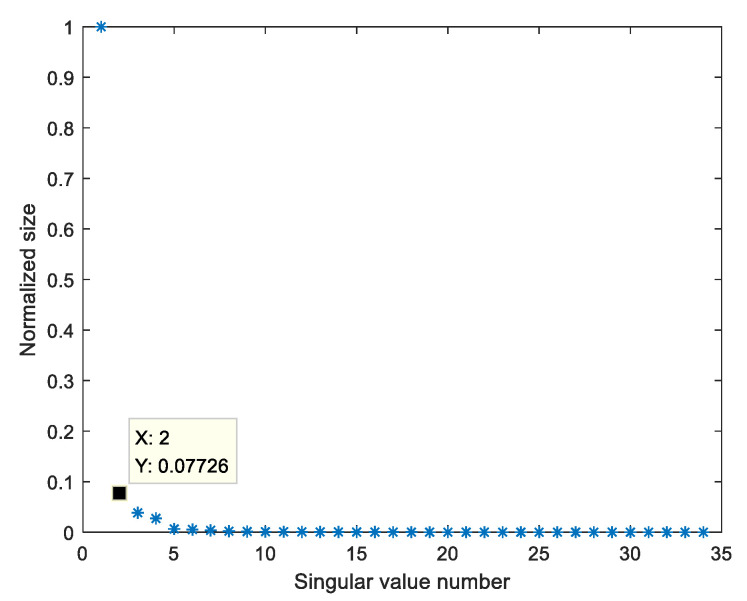
Singular value spectrum of radar echo.

**Figure 6 sensors-23-00756-f006:**
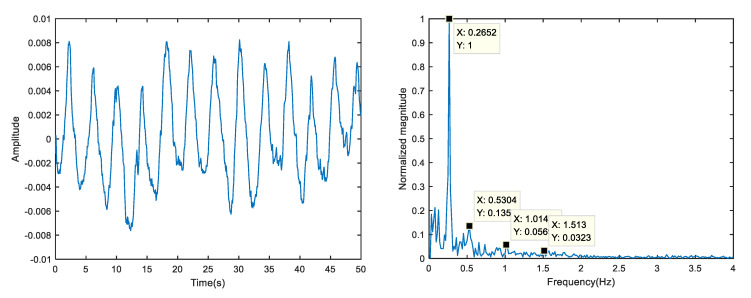
The original vital signs signal time domain and spectrum of target.

**Figure 7 sensors-23-00756-f007:**
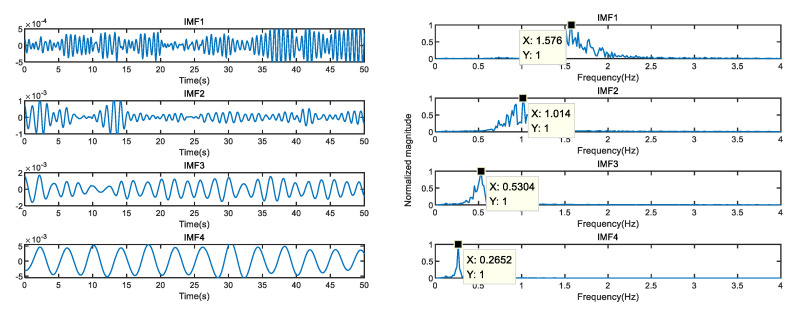
The original vital signs signal time domain and spectrum of target.

**Figure 8 sensors-23-00756-f008:**
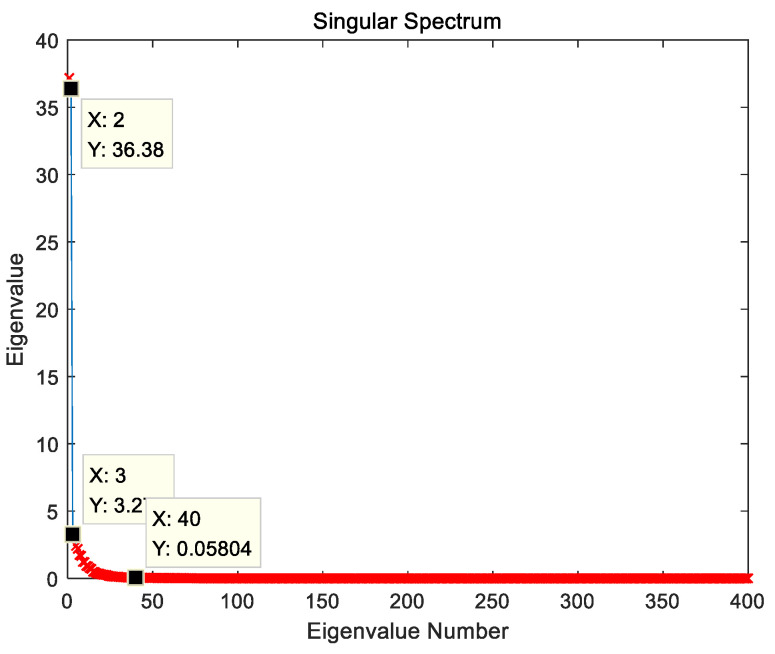
Singular value spectrum of trajectory matrix.

**Figure 9 sensors-23-00756-f009:**
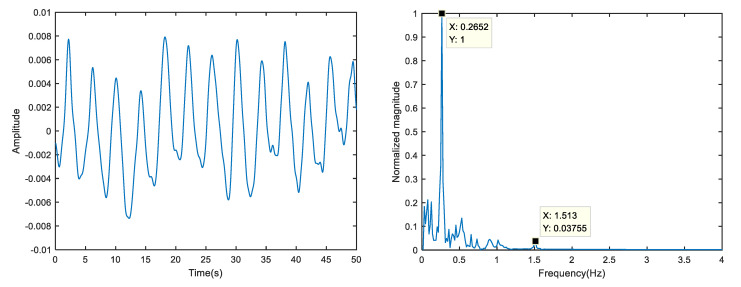
SSA reconstruct vital sign.

**Figure 10 sensors-23-00756-f010:**
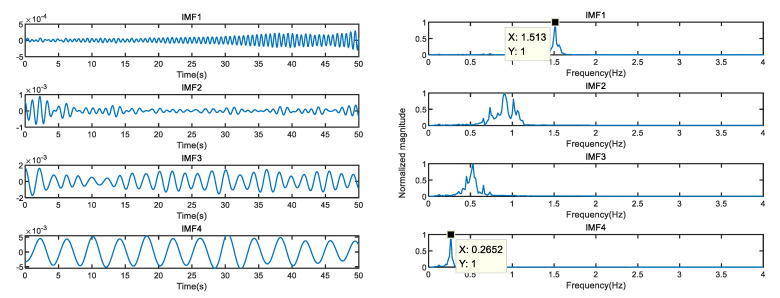
SSA-VMD result.

**Table 1 sensors-23-00756-t001:** The X4M03 module parameter setting.

Parameter	Value
Signal Pattern	Pulse
Center Frequency	7.290 GHz
Pulse Repetition Frequency	15.2 MHz
Test Time	50 s
Test Range	1.5 m
Output Type	Baseband

**Table 2 sensors-23-00756-t002:** Parameter of VMD algorithm.

Parameter	Value
***k*,** the number of intrinsic modes	5
τ, DUAL ASCENDING STEP LENGTH	0
DC, True if the first mode put at zero frequency	1
INIT, INITIALIZING THE CENTER FREQUENCY	0
ε, CONDITION OF CONVERGENCE	$10^{-6}$
α, the convergence rate	1000

**Table 3 sensors-23-00756-t003:** Experimental comparison results at 0.6 m.

Object	Standard HR/Hz	VMD	Error/Hz	SSA-VMD	Error/Hz
A	1.400	1.357	−0.043	1.357	−0.043
B	1.250	1.513	+0.263	1.217	−0.033
C	1.150	1.232	+0.082	1.186	+0.036
D	1.133	1.061	−0.072	1.061	−0.072
E	1.383	1.498	+0.115	1.342	−0.041

**Table 4 sensors-23-00756-t004:** Experimental comparison results at 1.2 m.

Object	Standard HR/Hz	VMD	Error/Hz	SSA-VMA	Error/Hz
F	1.367	1.310	−0.057	1.310	−0.057
G	1.500	1.560	+0.060	1.560	+0.060
H	1.050	1.232	+0.182	1.030	−0.020
I	1.400	1.435	+0.035	1.435	+0.035
J	1.517	1.482	−0.035	1.482	−0.035

## Data Availability

The data presented in this study are available on request from (W.H).

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
