# Peer review of "SSA-VMD for UWB Radar Sensor Vital Sign Extraction"

_sensors, 2023, doi:10.3390/s23020756_

Round 1

Reviewer 1 Report (New Reviewer)

some line in figures is too thin,make more clear.

Author Response

Thank you for taking time out of your busy schedule to read and give us valuable suggestions on the improvement of our manuscript! I wish you all the best! Our Cover letter is attached, please check!

Reviewer 2 Report (New Reviewer)

You are presenting a specific processing method applied to UWB radar data (SSA-VMD), and claim that this method results in improved SNR and therefore improved heart rate measurement accuracy. You have supported these claims with a set of figures that aim to illustrate the benefit if the method, as well as a quantitative comparison of HR accuracy between the proposed method and a more commonly used ‘reference method’ (VMD).

The theory of the dataset and algorithms are well described (although some aspects appear to be missing). It is good to see an attempt at validation on human subject data. The work is lacking however in the following aspects:

  • You claim improvement to SNR, which supposedly is the main advantage of your proposed algorithm. However, you did not quantify this. You only support this with a visual comparison of Fig 7 to Fig 9. This is not sufficient
  • You claim improvement in accuracy of HR measurement. This is claim is not supported: As anyone who has worked on this problem will know, the challenge with radar-based heart rate monitoring (especially in real word environments on a wide range of users) is selecting which of the spectral peaks are HR, which are respiratory rate (RR), which are HR or RR harmonics, and which are intermodulation between any of those, and which are caused by motion artifacts. You have completely avoided this problem, by manually selecting the HR harmonic, based on your knowledge of the reference device. This is completely unacceptable. If you want to claim that your algorithm leads to better HR measurement, then you must solve this problem, and describe how HR was obtained from your data.
  • Finally, it is unclear where exactly the novelty is. Your introduction does not provide an extensive literature review of the use of VMD and SSA for this problem. A quick google search teaches that VMD and SSA has been used before in radar-based vital sign monitoring. Because you did not describe the novelty in this work, and it is not apparent.

Below are a list of specific items that must be addressed:

  • Title is too generic. You are not the first to publish UWB-based vital sign monitoring. Title should be descriptive of the novelty in the work.
  • This manuscript has nothing to do at all with the power grid. Methods are not specific to the problem described in the introduction. Nothing in your methods or results suggests that you could possibly monitor vital signs of a grid employee in a real world scenario, or ‘avoid accidents’. You should reframe the problem entirely. What you are proposing is a specific combination of algorithms applied to UWB radar. Discuss the different methods, their advantages/disadvantages, and how this work compares to published work
  • At times language is unclear. Do a proper proofread.
  • line 49, reference 6: wrong reference. Has nothing to do with BCG
  • line 49: BCG is not a contactless method per se. it is employment of physiological phenomenon for HR measurement. This can be done in a contact manner (eg piezo) or contactless manner, using eg. radar or video. However, in your argument funnel, you are trying to present BCG as an alternate sensor modality to radar, which it isnt. Also not correct to generally state that SNR of BCG is low without explaining yourself.
  • refs 7-11. references appear messed up. Use a reference manager
  • Fig 1: not very helpful for those unfamiliar with radar-based vital sign monitoring. Explain how this relates to human vital signs. Most readers are not aware that ventricular contraction causes chest displacement.
  • line 90: delta indicates a dirac delta? Specify.
  • Figure 2 is lacking. I would like to see a full schematic of the algorithm used, from raw sensor data to HR value.
  • you describe you are working with complex data and that real world data has DC offset. How did you process complex IQ data? What assumptions were made? 
  • Line 160: This is really only valid under no-motion conditions, which is not what your introduction describes. You simply select the most ‘energetic’ range bin. This will certainly fail in a real world scenario where your participants aren’t instructed to sit still
  • line 197 Grouping. How do you determine which are ‘useful’ and which aren’t? What is the value of r and how is it determined? You currently describe this in Section 4 ‘experimental results’, which leads us to believe that your processing parameters are results as opposed to methods.
  • Section 4.1 and 4.2 have the same title..
  • what is a RocSea finger clip type device and what does it do
  • In section 4.3 you describe how some algorithmic parameters were chosen. Do you think these will be valid outside of this experiment? How would an algorithm like this function in real world environment?
  • Figure 9: demonstrated that the SSA method renders a cleaner spectrum than VMD only. But does not address the main challenge in HR monitoring: how do you select HR from this
  • line 347: average error? do you mean ‘mean absolute error’? or ‘mean error’?
  • How did you select HR? from all description it appears you select manually, based on proximity to reference. not acceptable
  • Line 364, figure 14h and j. It is obvious from these figures that you use unsuitable demodulation methods of complex IQ data. Doubling of respiratory waveform oscillation is likely a result of improperly applied arctangent demodulation. Describe your methods.
  • Table 4: results are identical between both compared methods, for all but 1 participant. Surely you cannot claim that one method is better than the other if they render almost identical results
  • abstract: first half is irrelevant. paper has nothing to do with grid safety and the methods or experiment do not support any claims about real world applicability of the proposed methods
  • abstract: you state that SNR is improved. Document does not support this with evidence other than a visual comparison of two anecdotal figures. If this is the result you wish to highlight, then quantify it
  • abstract: last claim: not supported
  • Introduction presents SSA as a novelty, but does not appear to be. SSA for UWB radar-based vital sign monitoring has been published previously, for example:
    • DOI:10.1109/JSEN.2019.2962721
    • DOI: 10.1109/PIERS.2016.7735605

Also multivariate SSA:

  • https://doi.org/10.1016/j.sna.2020.111968
  •  

Author Response

Thank you for taking time out of your busy schedule to read and give us valuable suggestions on the improvement of our manuscript! I wish you all the best! Our Cover letter is attached, please check!

Reviewer 3 Report (New Reviewer)

Specific and Thorough Comments to the Author
Comments:

       1. In lines 184 and 185, the overall effect of formula embedding needs attention.

2. In line 188, please check the sentence “Equation (6) is the Hankel matrix”.

3. In line 208, I recommend that the author briefly describe Figure 5.

4. In the experimental example given by the author, as shown in Figure 9, the Fourier spectrum of the signal after SSA has already had some effect. Has the author considered the effect of direct filtering?

5. I suggest that the authors include the effects of breathing signal extraction in the experiment. Do more experiments.

6. It is advised that Figure 14 be replaced with other graphics, such as a comparison of the separation effects among methods.

Author Response

Thank you for taking time out of your busy schedule to read and give us valuable suggestions on the improvement of our manuscript! I wish you all the best! Our Cover letter is attached, please check!

Reviewer 4 Report (New Reviewer)

The authors propose an approach which combines singular spectrum analysis (SSA) and variational model decomposition (VMD) for vital signs detection (heart beating and respiration) based on UWB radar signals. The paper presents nice ideas, however it could be improved prior to publication.

-In the state-of-art part, in the introduction, it would be relevant to add these two papers which deal with vital signs monitoring :

[1] Minhhuy Le, Dang-Khanh Le, Jinyi Lee, Multivariate singular spectral analysis for heartbeat extraction in remote sensing of uwb impulse radar,
Sensors and Actuators A: Physical, Volume 306, 2020,111968, ISSN 0924-4247, https://doi.org/10.1016/j.sna.2020.111968.

[2] Khan, F.; Azou, S.; Youssef, R.; Morel, P.; Radoi, E. IR-UWB Radar-Based Robust Heart Rate Detection Using a Deep Learning Technique Intended for Vehicular Applications. Electronics 2022, 11, 2505. https://doi.org/10.3390/electronics11162505

-Sentence on line 214 is not clear, "For the selection of the number of reconstructed singular values r, since the reconstructed object
is a sign signal, it is effective not to lose the weak heartbeat information
". Please clarify it and clarify the choice of the singular values r.

-In section 4.1, the authors cite reference [20] which is not included in the references list. Please add it to the references list.

-In section 4.1, line 188, It is not sure that the equation (6) is the Hankel matrix, please verify if the correct equation is cited.

-In line 207, in variable x_i+l, is it l or L?

-Please verify equation (23) : replace small l by L and also there are two symbols "=" which should be eliminated.

-Section 4.2 has the same title as section 4.1. Please correct the title, it is more about VMD rather than SSA.

-It might be interesting to add the VMD algorithm, which parameters are defined in Table 2.

-Please change the number of section "Experimental results analysis" (replace 4 by 5) and also the subsections numbers.

-In the subsection "Target Vital Sign Signal Extraction", please clarify the sentence "Import the radar data into the PC, picture show that the radar raw echo matrix composition is complex". Improve the form.

-In the same subsection, Figure 5 is not cited.

-In section "comparison of VMD and SSA-VMD", variable K in Table 2 must be in small case (k), as in line 288.

-In line 288, please remove "res".

-It might be interesting to show results for different values of window L.

-Please justify the number of singular values (0.1L).

-Figures 14 are not relevant, they may be concatenated in one figure for all targets.

-It is not clear how the authors deal with the fact that breathing harmonics may occur in the frequency range of heart rate? How to distinguish between harmonics and HR.

Author Response

Thank you for taking time out of your busy schedule to read and give us valuable suggestions on the improvement of our manuscript! I wish you all the best! Our Cover letter is attached, please check!

Round 2

Reviewer 3 Report (New Reviewer)

  • The author gave a detailed answer to my question.

This manuscript is a resubmission of an earlier submission. The following is a list of the peer review reports and author responses from that submission.

Round 1

Reviewer 1 Report

This paper presents a non-contact, radar-based vital sign monitoring approach. The paper presents an innovative processing step that provides heart rate detection accuracy of over 90% with analysis performed on 10 subjects.

Major Comments

Although the processing step appears to be novel, the way information/data is presented in the paper is confusing and major steps in the methodology section are missing (e.g., details about how the notch filter was implemented). Further, the background section does not mention other non-contact vital signs monitoring technologies besides radar-based systems – for instance, there is no mention of unobtrusive systems (e.g., ballistocardiography-based). It also appeared, due to the flow of the paper, that analysis was only performed on one subject until page 15 – and even with the additional subjects, most had higher than normal resting heart rates (60 to 70 beats per minute), making it unclear how these results would generalize. Thus, the paper needs to detail the limitations of the study. The lowest heart rate for the 50-second window was 1.383 Hz or roughly 83 beats per minute. Another major concern is that the paper discusses simulation results, but no information about the simulation was provided – e.g., how was the simulation data generated?

Minor Comments

Several statements in the introduction seem out of place. For example, the paper states on lines 35, 36, and 37, “In addition, in the fields of post-earthquake search and rescue, border crossing detection, and indoor living orientation, the human body’s respiration and heartbeat provide the basis for the existence of life.” There are several places where the paper states, “etc.” These areas need more details and references to improve the clarity of the paper. Was there institutional board (IRB) oversight? If so, details for the IRB should be provided. 

Reviewer 2 Report

1.       Authors should make a good layout according to the template provided by the journal and try to avoid content separation, such as in Table 1, Figure 11, Table 3, Table 4, and Figure 14.

2.       The author lists 40 articles in the bibliography, but only 23 articles are cited in the text. Where are the remaining 17 papers cited?

3.       Equations (27) and (28) appear to be incomplete.

4.       All figures in this paper should be presented in a suitable format and high resolutions. Significantly, the fonts in each figure should be uniform and precise.

5.       In the abstract, Some abbreviations should be given their full names when they first appear, like “CEEMD” and “SVD-K-MEANs.”

6.      In line 87, The word "draw" has to be replaced by " drawn" please improve the English writing carefully. The sentences in lines 479 and 480 may make the reader confused.

7.       The authors claim that the SVD is used for clutter suppression, but in my view, the authors should give a clear demonstration or some reference paper for choosing the suitable singular values. Meanwhile, the authors should carefully modify this manuscript; the equation (16) and (13) should use a correct denotation, such as in line 180, the should be instead by .

8.       The proposed method is validated in a single experimental scenery in this paper. We suggest that the author should add other experimental scenery to test the performance of the proposed method.

9.       In line 496, the authors claim that the proposed method “which can obtain real-time separated respiration and heartbeat signals.” We suggest that the author should make some demonstration on the real-time running.

10.    In lines 193, 194, and 195.  the denotation of the equations should be clarified. Such as , , and .

11.    In line 202, “It does not have the problem of modal aliasing in the EMD algorithm, and compared with the EEMD algorithm, the time-consuming and reconstruction errors are small”. Is it really? Even in the VMD algorithm, modal aliasing also exists.

12.    The results presented in Figure 10 are mismatched. Moreover, the author should tune the parameter of VMD. Too many papers show the performance of the VMD better than the EEMD-type algorithm.

13.    The innovation of this paper is insufficient. The authors should make a summary of the innovation in this paper.